# The Effect of Various Types of Biochar Mixed with Mineral Fertilization on the Development and Ionome of Winter Wheat (*Triticum aestivum* L.) Seedlings and Soil Properties in a Pot Experiment

**Grzegorz Kulczycki** [1], **Elżbieta G. Magnucka** [2], **Małgorzata P. Oksińska** [2], **Jolanta Kucińska** [2], **Rafał Kobyłecki** [3], **Katarzyna Pawęska** [4], **Robert Zarzycki** [3], **Andrzej Kacprzak** [3] and **Stanisław J. Pietr** [1,*]

[1] Department of Plant Nutrition, Wrocław University of Environmental & Life Sciences, Grunwaldzka 53, 50-357 Wrocław, Poland; grzegorz.kulczycki@upwr.edu.pl

[2] Agricultural Microbiology Lab, Department of Plant Protection, Wrocław University of Environmental & Life Sciences, Grunwaldzka 53, 50-357 Wrocław, Poland; elzbieta.magnucka@upwr.edu.pl (E.G.M.); malgorzata.oksinska@upwr.edu.pl (M.P.O.); jolanta.kucinska@upwr.edu.pl (J.K.)

[3] Department of Advanced Energy Technologies, Częstochowa University of Technology, Dąbrowskiego 69, 42-201 Częstochowa, Poland; rafal.kobylecki@pcz.pl (R.K.); robert.zarzycki@pcz.pl (R.Z.); andrzej.kacprzak@pcz.pl (A.K.)

[4] Institute of Environmental Engineering, Wrocław University of Environmental & Life Sciences, Grunwaldzki Square 24, 50-363 Wrocław, Poland; katarzyna.paweska@upwr.edu.pl

[*] Correspondence: stanislaw.pietr@upwr.edu.pl; Tel.: +48-71-320-5612

**Abstract:** This paper focuses on the agronomic evaluation of a synthetic NPK (N in the form of urea, P and K in the form of phosphate monopotassium) fertilizers blended with four types of pine (*Pinus sylvestris* L.) wood biochar prepared at different thermal regimes (300 °C, 400 °C, 600 °C and 700 °C). The evaluation of benefits was done based on crop nutritional status and soil fertility. The pot experiment was set up with fertile Haplic Luvisol fertilized with 1.85 g kg$^{-1}$ of blends of biochar (1.25 g) with urea (310 mg) and $KH_2PO_4$ (290 mg), which is equivalent to 500 kg ha$^{-1}$ (biochar ~67.6%; N ~7.8%; P ~3.6%; K ~4.7%) applied before sowing. Only NPK blends made with biochar containing 75% or 85% carbon increased the biomass of 27-day old wheat seedlings from 12% to 20% in comparison to NPK applied alone. These blends raised the content of Mn and Fe in plants but decreased the contents of Ca and Mg. All the tested mixtures enhanced soil fertility by increasing the content of humic acids. Additionally, the content of potentially phytotoxic phenolic compounds was lower. In general, the addition of biochar to NPK fertilizer did not show a negative effect on crop quality. The overall results of the study suggest that the application of low doses of biochar to synthetic fertilizer can benefit crops and can support soil fertility.

**Keywords:** NPK fertilizer; blends; soil organic matter; nutrients uptake; humic acids; phenolic compounds

## 1. Introduction

The application of different forms of organic material is a well-known, traditional strategy used to recover soil fertility by replenishing soil organic matter (SOM), and consequently, stimulating the development, resistance, and nutrient profile (ionome) of plants. However, the necessity of using bulk quantities of traditional manure, which has a short life span in soil, limits its effectiveness. Fertilizing with

biochar (BC), the product of pyrolysis of organic materials in a low or no oxygen environment (see Antal and Grønli [1]) has become an interesting option for agricultural production. Applying biochar at high doses can improve soil fertility in different climate zones. The yield-stimulating effect of biochar is especially beneficial for crop production in the low-nutrient, acidic soils in the tropics [2–8]. In addition, applying biochar to the soil is a potential strategy for carbon sequestration to mitigate climate change [9,10]. A high proportion of biochar can also increase soil water retention [11]. This effect has especially been seen in sandy soil enriched with biochar prepared at higher pyrolysis temperatures due to its high surface area [12,13] and indirectly via subsequent increases in organic carbon (C) in the soil [14] when plant growth was stimulated. The contradicting positive and negative effects of applying BC on SOM fractions have been described in the literature. Among others, Cross and Sohi [15] and Steiner et al. [16] reported that BC can alter the mineralization of organic matter in the soil. This fact is linked to the microbial release of nutrients such as nitrogen (N) from the SOM to compensate for high C:N ratios after the use of BC [17,18]. Alternatively, the added organic matter from plant residues was incorporated more rapidly into stable organic mineral fractions of BC-rich as opposed to BC-poor soils from the Central Brazilian Amazon [19]. Furthermore, total C mineralization was lower in BC-rich soils, despite a higher microbial biomass than in BC-poor adjacent soils during almost 1.5 years of incubation [19]. Demisie et al. [20], Lin et al. [21] and Tian et al. [18] also reported increases in the microbial biomass of C, the dissolved organic C content (DOC) and the level of the light fraction of organic C in BC-enriched subtropical soils compared with soils unamended with BC. In other field studies of temperate zone soils, the application of BC did not affect levels of either dissolved organic N (DON) or carbon [22], whereas it decreased the DOC concentration in Chernozem [23]. These contradictory effects of applying BC on SOM fractions may be attributed to the specific processes governing C and N cycling under specific climatic conditions, as well as to varying management practices such as adding NPK fertilizer or other plant residues. Moreover, there is a lack of comparative research on the effects of different qualities of BC produced under various temperature regimes from the same type of organic substrate, in particular with respect to the content of C and N, which vary depending on production temperatures [24]. Elzobair et al. [25] demonstrated that the presence of biochar did not affect the activities of β-glucosidase, β-D-cellobiosidase or *N*-acetyl-β-glucosaminidase. In turn, the activity of β-xylosidase, which is essential for the complete breakdown of xylans, was markedly decreased. Other authors also noted reduced enzyme activities in biochar-enriched soil [26,27]. Still, others showed the positive impact of biochar on enzymatic activity in the soil [18,28–31]. The varied effects were probably dependent on both the type and dose of biochar applied, which may impact the ability to sorb organic compounds and not just enzymes [25,29,30]. The aforementioned studies were mostly done with pure BC applied at high doses $\geq 10$ Mg ha$^{-1}$ under field conditions or $\geq 2.5\%$ (*w/w*) in pot experiments. The use of such high doses of biochar in agricultural practice on large areas for main cash crops such as cereals, maize or potato is technically difficult. Moreover, the present market price of BC offered by firms targeting industrial agricultural application ranges from \$300–500 Mg$^{-1}$ in the US [32], and in Europe, according to Schmidt and Shackley [33], the cost of biochar was found to be as low as €200 Mg$^{-1}$. Currently, these obstacles make it possible to use BC only in small quantities or in high-end specialty markets. One option for introducing BC into agricultural practice is to combine the application with NPK fertilizers, as a proposed slow-release fertilizer to enhance soil fertility [34–38] or use as an alternative planting substrate to replace the conventional black peat [39]. It is also worth emphasizing that there have been very few comparative studies on the impact of applying NPK fertilization with biochar produced from pine wood chips either on agricultural crops or on SOM. Among different organic substrates used for biochar production, the most suitable materials are pine wood chips from forests in temperate regions of the world. Winter wheat was used for the phytotron bioassay, because it is the main cereal crop in Central Europe and is grown on good arable soils. Interestingly, it was hypothesized that the various levels of biochar carbonization used as soil amendment led to different biological activity in the soil, SOM composition, and nutrient status of plants during short-term incubation. Thus, the aim

of this study was to investigate in a pot experiment the impact of low doses of four types of biochar prepared under different thermal regimes together with NPK fertilization on the growth of winter wheat seedlings, the mineral uptake by the plants, the changes in the quality and quantity of soil organic matter and the changes in some of the enzymatic activities in the soil.

## 2. Materials and Methods

### 2.1. Tested Types of Biochar

Four types of biochar were prepared from pine (*Pinus sylvestris* L.) wood chips under different thermal regimes in an electrically-heated furnace in an oxygen-free atmosphere until the end of the devolatilization process. Biochar type I, biochar type II, biochar type III and biochar type IV were prepared by the devolatilization of pine wood chips at 300 °C for 70 min, at 400 °C for 20 min, at 600 °C for 8 min and at 700 °C for 7 min, respectively. The porosity determined by a mercury porosimeter (Quantachrome Instruments model PoreMaster 33, Anton Paar QuantaTec, Inc., Boynton Beach, FL, USA) of tested biochars were 31.5%, 53.5%, 46.2% and 26.4%, respectively. The chemical characteristics of the types of biochar tested are summarized in Table 1.

**Table 1.** Chemical properties of tested types of biochar used in a pot experiment.

| Type of Biochar | Abbreviations * | H (%) | C (%) | N (%) | P (%) | K (%) | S (%) |
|---|---|---|---|---|---|---|---|
| Biochar type I | BC52% | 5.4 a | 52 c | 0.39 a | 0.059 a | 0.321 a | 0.004 a |
| Biochar type II | BC50% | 4.3 a | 50 c | 0.33 b | 0.058 a | 0.301 a | 0.004 a |
| Biochar type III | BC75% | 2.6 b | 75 b | 0.29 c | 0.053 a | 0.290 a | 0.003 a |
| Biochar type IV | BC85% | 1.9 b | 85 a | 0.26 c | 0.054 a | 0.294 a | 0.003 a |

* The percentage values of the abbreviations refer to the C content of biochar. Values are the mean of three replicates of each sample. Values are followed by different letters in columns indicating significant differences according to Tukey's test ($p < 0.05$).

### 2.2. Pot Experiment

Caryopses of winter wheat (*Triticum aestivum* L.) cv. Scirocco were used in the pot experiment. The pots were filled with soil collected from the organic horizon of a commercial field in Przeworno (50°68′ N 17°18′ E), which was a Haplic Luvisol (loamic) soil (pH$_{KCl}$ 7.1, C$_{tot.}$ 0.99%). The soil was dried and sieved using a 2 mm sieve. The granulometric composition of the soil was as follow: sand 62% with dominant medium and fine fractions, silt 11% and clay 27%. The content of plant-available nutrients in the soil was as follow: phosphorus P 158 mg kg$^{-1}$, potassium K 162 mg kg$^{-1}$ and magnesium Mg 110 mg kg$^{-1}$. The soil had not been previously manured with biochar. The following groups were set up: CONTROL—control soil without fertilization; NPK—soil with NPK (24:11:14) fertilization composed of urea (310 mg kg$^{-1}$) and KH$_2$PO$_4$ (290 mg kg$^{-1}$); NPK + BC52%—soil with NPK fertilization as above mixed with biochar type I; NPK + BC50%—soil with NPK fertilization as above mixed with biochar type II; NPK + BC75%—soil with NPK fertilization as above mixed with biochar type III; NPK + BC85%—soil with NPK fertilization as above mixed with biochar type IV. All the tested types of biochar were applied with a dose of 1.25 g blended with 600 mg of NPK fertilizer, and the mixture in a dosage of 1.85 g kg$^{-1}$ of air-dried soil was carefully mixed. This was an equivalent to 500 kg ha$^{-1}$ of blend (biochar ~67.6%; N ~7.8%; P ~3.6%; K ~4.7%) applied before sowing. Each plastic pot (volume 1 dcm$^3$, height 8 cm; circumference 45 cm) was filled with 1 kg of tested soil and the soil layer was about 6 cm high. The space among pots was kept at a distance of ~5 cm. Each group was set up with three replications. The experiment was conducted as completely randomized designs after each watering in three replications. Pots were watered up to 60% of water holding capacity and stored for 3 weeks at 16–18 °C in a dark room before sowing. During the incubation period, the constant soil moisture based on weight loss was kept at 60% of the water holding capacity by adding deionized water every second day. Such a level of moisture secures a proper proportion of air–water phases. Thirty seeds of winter wheat were placed in each pot with the soils as described after 30 days of

incubation. Pots with plants were kept in a controlled growth chamber with a photoperiod 16 h/8 h light/dark and 26–28 °C/16–18 °C day/night temperatures for 27 days.

*2.3. Plant Analyses*

2.3.1. Study of Wheat Germination and Biomass

Observations of germination were conducted 5 and 9 days after sowing the seeds in the pots. The proportion of seeds that germinated in each soil amendment was calculated. The model provided a statistical test of the hypothesis that germination was not affected by the amendment, and the standard error for each proportion was calculated. After the plants germinated, the number of plants was equalized to 25 seedlings per pot. Tested plants were grown for 27 days and harvested. The fresh biomass of the tested winter wheat was dried at 105 °C and subjected to chemical analysis.

2.3.2. Chemical Plant Analysis

The levels of macronutrients, micronutrients and metals were determined in the dry mass of the harvested plants, which were ground and subjected to mineralization. The nitrogen content was determined using the Kjeldahl method, involving wet digestion and distillation [40]. The total sulfur content was determined with the Butters and Chenery method [41]. Dry mineralization was used to determine other macro- and micronutrients. The contents of the following elements were assayed in a mineralized solution: P—according to the colorimetry method, K and Ca by the flame photometry method. Mg, Mn, Fe, Cu, Zn, Ni, Cd and Pb were assayed using atomic absorption spectrometry (AAS) (Varian model SpectrAA 220FS, Varian Medical Systems, Inc., Charlottesville, VA, USA).

*2.4. Soil Analysis*

2.4.1. Preparation of Soil Samples

After cutting off the above-ground parts of winter wheat, bulk soil samples were immediately frozen in liquid nitrogen and then lyophilized for enzymatic analysis. The lyophilized soil samples were stored at −72 °C. After removing the roots, the remaining bulk soil samples were dried (110 °C), sieved (2.0 mm) and stored at room temperature for later use.

2.4.2. Physicochemical Properties of Soil

Soil acidity was determined in 1:2.5 soil:1 M KCl suspensions using a digital pH meter CP 505 (Elemetron Co., Zabrze, Poland). The total content of C, N and S in the tested samples was determined by the Dumas method of combustion using a TruSpec analyzer (Leco, Co., St. Joseph, MI, USA). The product gas, containing $CO_2$, $H_2O$, $SO_x$ and $NO_x$, was then passed through a series of infrared detectors to determine the amount of C and S in the sample. The amount of N was determined with the use of a thermal conductivity detector. Prior to the analysis, the sample gas was swept through hot copper to remove oxygen and change NOx to N, and then with Lecosorb and Anhydrone absorbents to remove carbon dioxide and water. The content of plant-available phosphorus and potassium was determined with the Egner-Riehm DL method [42] and the content of soluble magnesium with the Schachtschabel method [43]. The content of soluble micronutrients and heavy metals in the tested soils, such as manganese (Mn), iron (Fe), copper (Cu), zinc (Zn), nickel (Ni), cadmium (Cd), lead (Pb) and chromium (Cr), was determined with the Rinkis method [44] using AAS (Varian model SpectrAA 220FS, Varian Medical Systems, Inc., Charlottesville, VA, USA).

### 2.4.3. Soil Organic Carbon

Soil Humic Acids

Soil humic acids (HA) were extracted from 30 g of dried soil with the modified Swift [45]. After shaking with 0.1 M NaOH (1:10, *w/v*) for 24 h, the soil sample was centrifuged (2500× *g*, 24 °C, 20 min). The supernatant was combined with the following supernatants obtained by first washing soil pellets with 0.1 M NaOH and then with distilled water. Directly afterwards, the whole liquid was acidified to pH 1.0 with 12 M HCl and left to precipitate the insoluble HAs for another 24 h using a rotatory shaker at 150 rpm. The decanted pellet of HAs after washing with 0.1 M HCl was suspended in 0.1 M NaOH for 1 h and then washed with distilled water again. All supernatants separated from the residue by centrifugation were joined and acidified to pH 1.0 with 0.1 M HCl. Black colored HAs were precipitated by centrifugation and also washed with distilled water. The obtained sediment was dissolved overnight in 20 mL of 0.02 M $NaHCO_3$. Finally, the suspension of HAs was precipitated by acidification with a mixture of HCl/HF acids (100:1, *v/v*) to pH 1.0. After 24 h, the pellet of HAs was collected by centrifugation and washed several times with distilled water until the reaction on the Cl-ions disappeared. The obtained precipitate was dried under a vacuum until the constant weight and dry mass per 1 kg of tested soil sample was calculated.

Soil Free Phenolic Acids

The extraction procedure of free phenolic acids in the tested soil samples was based on the Krygier et al. [46] with some modification. Free phenolic compounds were extracted from 50 g soil samples five times with 100 mL of acetone/methanol/water (7:7:6, *v/v*). The combined supernatants were filtered and separated by centrifugation (10 min, 1000× *g*) at a temperature < 30 °C. The organic fraction of the supernatant was evaporated under a vacuum at 45 °C, and the remaining water fraction was acidified with 6 M HCl to pH 2.0. The precipitate was removed by centrifugation as above. Then, the supernatant was extracted six times with hexane equal to the water phase and then six times with a mixture of diethyl ether/ethyl acetate (1:1, *v/v*). The combined organic fraction of diethyl ether/ethyl acetate was dehydrated with $Na_2SO_4$. After filtration, the solution of diethyl ether/ethyl acetate was evaporated under a vacuum at 30 °C until dry. The residue was dissolved in 2 mL of ethyl acetate. Absorbance was determined at 320 nm using ferulic acid (Sigma-Aldrich, Inc., Saint Louis, MO, USA) as a standard. The concentration of phenolic compounds was expressed as μg per 1 g of dry soil.

Glomalin Concentration

The total glomalin (TG) from the soil samples was extracted in a 50 mM citrate buffer, pH 8.0, according to Gałązka [47] with some modifications. The soil samples (10 g) were covered with the buffer and autoclaved at 121 °C for 60 min. Extraction was carried out several times until the organic fraction was totally washed out of the soil. After each autoclaving, the buffer containing the solubilized glomalin was poured off, and the soil samples were covered with the sterile buffer again. The buffer extractants, collected after each heating, were combined and supplemented at an equal volume for each sample and centrifuged at 10,000× *g* for 10 min at 4 °C. The supernatants were stored at 4 °C for later analysis. The glomalin content in the buffer extractants was determined according to Bradford [48] at 595 nm using bovine serum albumin (Sigma-Aldrich, Inc., Saint Louis, MO, USA) as a standard.

Water Extractable Carbon

The level of water extractable C was determined in fresh soil samples modified by Ghani et al. [49] using the method of Haynes and Francis [50]. The extraction of water extractable C was conducted in two steps. The first step involved the removal of readily soluble C from the soil that may have come from recent liming of the soil or from animal excreta or soluble plant residues. Soil samples equivalent to 5 g (oven-dry weight) were weighed into 50 mL polypropylene centrifuge tubes. These were extracted

with 50 mL of distilled water for 30 min on an end-over-end shaker at ~30 rpm and centrifuged at 20 °C for 20 min at 1000× *g* at 20 °C. Then, the supernatant was filtered (0.45 mm cellulose nitrate membrane) into separate vials for carbon analysis. The fraction of the soil organic carbon was classified as water soluble C (WSC). The second step involved the extraction of labile components of soil carbon at 80 °C for 16 h. This is subsequently referred to as hot-water extractable carbon (HWC). A further 50 mL of distilled water was added to the sediments in the same tubes. The tubes were shaken on a vortex shaker for 10 s to suspend the soil in the water. The tubes were capped and left for 16 h in a hot-water bath at 80 °C. At the end of the extraction period, each tube was shaken for 10 s on a vortex shaker to ensure that HWC released from the SOM was fully suspended in the extraction medium. The tubes were then centrifuged for 20 min at 1000× *g* at 20 °C. The supernatants were filtered (0.45 mm cellulose nitrate membrane). The total organic carbon (TOC) content in both fractions was measured using the Sievers InnovOx Laboratory TOC analyzer (GE Analytical Instruments, General Electric, Co. Boston, MA, USA). The content of water soluble organic carbon (WSOC) and of the hot water soluble organic carbon (HWSOC) in the tested samples was estimated as the difference between the content of total carbon (TC) and inorganic carbon (IC) of WSC and HWC, respectively. A measurement of each sample was made in four repetitions with a flush of dilution water after each analysis. Before starting the analysis, the pH of each sample was evaluated in order to determine the volume of acid and oxidizer needed for measurement. Hydrochloric acid 3 M (HCl) at 5% of the sample volume and an oxidizer (sodium persulfate $Na_2S_2O_8$) at 15% of the sample volume was used. The aim of using the acid was to reduce the pH to 2 to enable the transformation of carbonate salts or bicarbonates to carbon dioxide. Then, the mixture was pumped to the reactor at 375 °C and under 22.1 MPa pressure. The inorganic C content in the extracts was generally less than 4% of the total hot-water extractable C.

Enzymatic Activity

Lyophilized soil samples were used to estimate soil enzyme activities with the method described by Schinner and von Mersi [51]. The enzyme activity was measured after incubating 1 g of toluene-treated soil (0.5 mL) with 3 mL of 2% (*w/v*) suspension of the respective substrate, i.e., carboxymethyl cellulose sodium salt (Sigma-Aldrich, Inc., Saint Louis, USA) or xylan from birch wood (Fluka Chemie GmbH, Buchs, Switzerland) prepared in a 2 M sodium acetate buffer (pH 5.5) at 40 °C. The resulting Prussian blue was measured at 690 nm. One unit of CMC-ase and Xylanase activity was defined as the amount of enzyme that released 1 nmol of reducing sugars as glucose equivalents per hour in 1 g of lyophilized soil. All enzyme activity was determined using a calibration curve for D-glucose.

## 2.5. Statistical Analysis

All tests were done in three replications. A variance analysis with a single classification was carried out and significant differences among the means were revealed through Tukey's test with a 95% level of significance [52].

## 3. Results

### 3.1. Germination and Yield of Winter Wheat Seedlings

Wheat germination was not noticeably affected by most of the tested treatments. After 5 days, the emergence capacities in the control soil without fertilization (CON) and with mineral fertilization (NPK) ranged from 84.3% to 87.8% and were significantly lower than in all the examined variants with biochar, in which the emergence capacity ranged from 93.3% to 94.4%. However, after 9 days, the germination capacity was not significantly different and ranged from 95.8% to 96.7% (data not shown). The application of NPK fertilization alone or with the four tested types of biochar significantly improved the yield of the fresh and dry biomass of seedlings in comparison to the control plants grown in unfertilized soil (Figure 1A,B). The application of BC52% or BC50% blended with NPK fertilization did not significantly change the growth of winter wheat seedlings in comparison with NPK

fertilization applied alone. However, blended NPK fertilization with BC85% significantly improved the yield of the fresh and dry biomass of wheat seedlings, by 12% and 20%, respectively (Figure 1A,B); additionally, the application of NPK fertilization blended with BC75% significantly improved the yield of dry biomass 20% in comparison with NPK fertilization alone (Figure 1B).

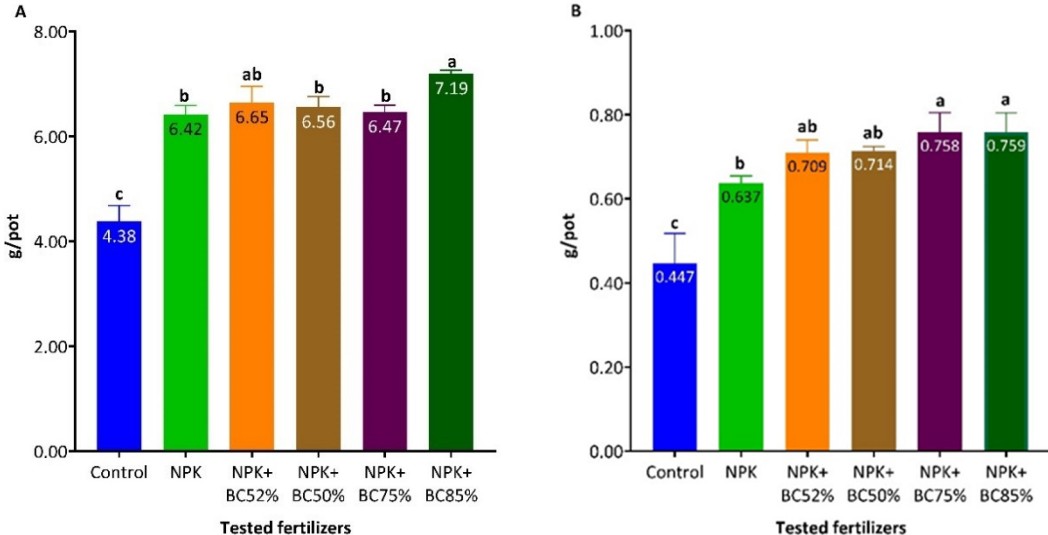

**Figure 1.** Average yield of above-ground (**A**)—fresh biomass, (**B**)—dry biomass. Treatments: control soil without fertilization (Control), soil fertilizer with NPK alone (NPK), soil fertilized with NPK and biochar type I containing 50% of C (NPK + BC52%), soil fertilized with NPK and biochar type II containing 52% of C (NPK + BC50%), soil fertilized with NPK and biochar type III containing 75% of C (NPK + BC75%) and soil fertilized with NPK and biochar containing 85% of C (NPK + BC85%). Different letters on the bars indicate significant differences according to Tukey's test ($p < 0.05$). Bars indicated the ±SD ($p < 0.05$).

### 3.2. The Amount of Macro- and Micronutrients and Heavy Metals in Winter Wheat Seedlings

As with the increase of the yield of the fresh and dry biomass of seedlings after the application of NPK fertilization alone, there was a significant increase in the content of macronutrients N, P, K and Mg but not of Ca in the seedlings of winter wheat in comparison to control plants grown in unfertilized soil (Table 2). However, the amounts of S were significantly lower after applying NPK alone or with any of the tested types of biochar. The application of the four tested types of biochar with NPK in most cases did not significantly change the content of N, P, K and S in seedlings compared to plants grown in soil fertilized with NPK alone (Table 2). However, noticeable decreases were observed in the amount of Mg after adding all four types of biochar and in Ca after applying BC50%, BC75% and BC85% types of biochar, in comparison with plants grown in soil with NPK alone.

**Table 2.** Amount of macronutrients in 27-day old winter wheat seedlings.

| Groups * | N tot. | P | K | S tot. | Mg | Ca |
|---|---|---|---|---|---|---|
| | g kg$^{-1}$ | | | | | |
| Control | 39.4 c | 6.92 c | 67.7 a | 4.79 a | 2.64 c | 12.0 ab |
| NPK | 45.5 ab | 8.01 ab | 61.7 b | 4.05 bc | 3.59 a | 13.9 a |
| NPK + BC52% | 48.4 a | 7.57 bc | 58.9 bc | 4.02 bc | 3.14 b | 11.5 ab |
| NPK + BC50% | 42.5 bc | 7.46 bc | 54.7 c | 3.68 c | 2.96 bc | 9.86 bc |
| NPK + BC75% | 42.1 bc | 8.11 ab | 58.5 bc | 3.84 bc | 2.80 bc | 9.56 bc |
| NPK + BC85% | 43.7 abc | 8.79 a | 58.9 bc | 4.16 b | 2.86 bc | 8.49 c |

* Control—soil without fertilization; NPK—soil fertilizer with NPK alone, NPK + BC52%—soil fertilized with NPK and biochar type I containing 50% of C; NPK + BC50%—soil fertilized with NPK and biochar type II containing 52% of C; NPK + BC75%—soil fertilized with NPK and biochar type III containing 75% of C; NPK + BC85%—soil fertilized with NPK and biochar containing 85% of C. Values are the mean of three replicates of each sample. Values followed by different letters in columns indicate significant differences according to Tukey's test ($p < 0.05$).

A comparison of the content of micronutrients and heavy metals in the seedlings of winter wheat are summarized in Table 3. The application of NPK alone noticeably decreased the content of micronutrients such as Fe and Cu as well as of heavy metals such as Cd and Pb, but increased the amount of Zn in the plant biomass compared to the unfertilized control plants. The combined applications of all types of biochar with NPK did not change the amount of Cu and Zn or heavy metals such as Ni, Cd and Pb in comparison with NPK fertilization applied alone. However, there was a noticeable increase in the concentration of Mn and Fe in the tested seedlings after using biochar BC75 and BC85 in comparison to all others tested plants (Table 3).

**Table 3.** Amount of micronutrients and heavy metals in 27-day old winter wheat seedlings.

| Groups * | Mn | Fe | Cu | Zn | Ni | Cd | Pb |
|---|---|---|---|---|---|---|---|
| | mg kg$^{-1}$ | | | | | | |
| Control | 67.3 ab | 938 b | 18.3 a | 64.2 b | 6.87 a | 1.08 a | 2.35 a |
| NPK | 59.3 b | 655 cd | 15.0 bc | 73.7 a | 5.38 a | 0.58 b | 1.09 b |
| NPK + BC52% | 55.0 b | 568 d | 14.6 bc | 72.7 a | 4.93 b | 0.67 b | 1.33 b |
| NPK + BC50% | 65.7 ab | 861 bc | 16.2 ab | 72.4 ab | 6.27 a | 0.62 b | 1.25 b |
| NPK + BC75% | 76.0 a | 1 419 a | 14.3 bc | 69.6 ab | 6.87 a | 0.56 b | 1.23 b |
| NPK + BC85% | 74.0 a | 1 425 a | 13.6 c | 72.3 ab | ND | 0.62 b | 1.07 b |

* Control—soil without fertilization; NPK—soil fertilizer with NPK alone; NPK + BC52%—soil fertilized with NPK and biochar type I containing 50% of C; NPK + BC50%—soil fertilized with NPK and biochar type II containing 52% of C; NPK + BC75%—soil fertilized with NPK and biochar type III containing 75% of C; NPK + BC85%—soil fertilized with NPK and biochar containing 85% of C. Values are the mean of three replicates of each sample. Values followed by different letters in columns indicate significant differences according to Tukey's test ($p < 0.05$).

### 3.3. The Physicochemical Parameters of Soil

The application of NPK fertilization alone or with the four tested types of biochar on the physical and chemical parameters of the soil after collecting 27-day old winter wheat seedlings did not significantly change soil acidity, the total content of carbon, nitrogen, sulfur or the amount of soluble magnesium in the soil in comparison with the unfertilized control groups (Table 4). The levels of plant available phosphorus were higher in all soil samples after NPK fertilization. Applying the tested types of biochar with NPK did not significantly change phosphorus availability in comparison to NPK fertilization applied alone. The amount of plant available potassium was higher in soil samples after applying NPK alone and after applying NPK with two types of biochar containing lower amounts of carbon, BC52% and BC50%, compared to the unfertilized control soil (Table 4). A noticeable decrease in plant available potassium was observed in the soil fertilized with NPK and biochar BC85%, which contained the highest amount of carbon (Table 4).

**Table 4.** The acidity and total amount of C, N and S, and amount of plant available macronutrients in the tested soils after harvesting 27-day old winter wheat seedlings.

| Groups * | $pH_{KCl}$ | $C_{tot}$ | $N_{tot}$ | $S_{tot}$ | $P_{pa}$ | $K_{pa}$ | $Mg_{sol}$ |
|---|---|---|---|---|---|---|---|
| | | \multicolumn mg kg⁻¹ | | | | | |
| Control | 6.99 a | 9810 a | 1313 a | 16 a | 184 b | 173 c | 90.2 a |
| NPK | 6.84 ab | 9520 a | 1003 a | 13 a | 225 a | 243 a | 92.1 a |
| NPK + BC52% | 6.70 b | 9750 a | 1293 a | 14 a | 238 a | 251 a | 91.6 a |
| NPK + BC50% | 6.77 ab | 9530 a | 1213 a | 11 a | 222 a | 225 ab | 92.2 a |
| NPK + BC75% | 6.81 ab | 9930 a | 1050 a | 13 a | 229 a | 213 abc | 95.4 a |
| NPK + BC85% | 6.79 ab | 9720 a | 1050 a | 11 a | 234 a | 192 bc | 94.3 a |

* Control—soil without fertilization; NPK—soil fertilizer with NPK alone; NPK + BC52%—soil fertilized with NPK and biochar type I containing 50% of C; NPK + BC50%—soil fertilized with NPK and biochar type II containing 52% of C; NPK + BC75%—soil fertilized with NPK and biochar type III containing 75% of C; NPK + BC85%—soil fertilized with NPK and biochar containing 85% of C; tot—total content; pa—plant available content; sol—soluble content. Values are the mean of three replicates of each sample. Values followed by different letters in columns indicate significant differences according to Tukey's test ($p < 0.05$).

For the tested fertilizers, NPK alone or NPK plus different types of biochar, there was no significant change in the amount of most soluble micronutrients and heavy metals in comparison to the unfertilized control soil samples after 27 days of winter wheat growth (Table 5). Among the tested micronutrients, only the amount of soluble Cu was significantly lower in soil samples from pots fertilized with NPK alone or in from pots fertilized with NPK plus two types of biochar (Table 5).

**Table 5.** Amount of soluble micronutrients and heavy metals in tested soils after harvesting 27-day old winter wheat seedlings.

| Groups * | Mn | Fe | Cu | Zn | Ni | Cd | Pb |
|---|---|---|---|---|---|---|---|
| | | | \multicolumn mg kg⁻¹ | | | | |
| Control | 148 a | 1185 a | 11.6 a | 14.58 a | 1.44 a | 0.15 a | 8.50 ab |
| NPK | 148 a | 1199 a | 7.16 bc | 20.07 a | 1.41 a | 0.15 a | 9.18 ab |
| NPK + BC52% | 152 a | 1219 a | 7.15 bc | 16.49 a | 1.45 a | 0.17 a | 9.61 a |
| NPK + BC50% | 144 a | 1148 a | 5.93 c | 15.25 a | 1.43 a | 0.16 a | 8.56 ab |
| NPK + BC75% | 151 a | 1166 a | 10.2 ab | 16.60 a | 1.41 a | 0.14 a | 7.90 b |
| NPK + BC85% | 138 a | 1101 a | 9.09 abc | 18.89 a | 1.41 a | 0.14 a | 8.24 ab |

* Control—soil without fertilization; NPK—soil fertilizer with NPK alone; NPK + BC52%—soil fertilized with NPK and biochar type I containing 50% of C; NPK + BC50%—soil fertilized with NPK and biochar type II containing 52% of C; NPK + BC75%—soil fertilized with NPK and biochar type III containing 75% of C; NPK + BC85%—soil fertilized with NPK and biochar containing 85% of C. Values are the mean of three replicates of each sample. Values followed by different letters in columns indicate significant differences according to Tukey's test ($p < 0.05$).

### 3.4. Soil Organic Carbon

Soil amendments with NPK alone did not change the levels of humic acids, free phenolic acids or glomalin, but significantly increased the content of WSOC, and at the same time significantly decreased the content of HWSOC in comparison with the unfertilized control soil (Table 6). The biochar applied together with NPK fertilization resulted in significant changes in the amount of most of the aforementioned soil organic carbon compounds, except for the content of HWSOC, in comparison with the control and NPK fertilized soils after 27 days of winter wheat cultivation. All four types of biochar significantly increased the humic acid content in soil samples ranging from 15.7% to 21.4%, while simultaneously decreasing the amount of free phenolic acids in the same soil samples ranging from 21.0% to 34.3% in comparison to soil fertilized with NPK alone. Moreover, adding the tested types of biochar, except for BC50%, to the NPK fertilization reduced the amount of glomalin's in the tested soils from 17.5% to 46.2% in comparison to soil fertilized with NPK alone. Only BC85% significantly lowered the amount of WSOC in comparison with soil fertilized with NPK alone, but it was still significantly higher than in the unfertilized control soil (Table 6).

**Table 6.** Number of organic carbon fractions in tested soils after harvesting 27-day old winter wheat seedlings.

| Groups * | Humic Acids | Free Phenolic Acids | Glomalin's | WSOC | HWSOC |
|---|---|---|---|---|---|
| | mg kg$^{-1}$ | | | | |
| Control | 3810 b | 1.428 a | 588.8 ab | 307 c | 1020 a |
| NPK | 4200 b | 1.431 a | 683.8 a | 443 a | 864 bc |
| NPK + BC52% | 5100 a | 0.975 cd | 317.0 b | 441 ab | 915 b |
| NPK + BC50% | 4860 a | 1.022 c | 450.9 ab | 427 ab | 858 bc |
| NPK + BC75% | 4990 a | 0.936 d | 322.4 b | 409 ab | 862 bc |
| NPK + BC85% | 4930 a | 1.112 b | 367.9 b | 400 b | 815 c |

* Control—soil without fertilization; NPK—soil fertilizer with NPK alone; NPK + BC52%—soil fertilized with NPK and biochar type I containing 50% of C; NPK+BC50%—soil fertilized with NPK and biochar type II containing 52% of C; NPK + BC75%—soil fertilized with NPK and biochar type III containing 75% of C; NPK + BC85%—soil fertilized with NPK and biochar containing 85% of C; WSOC—water soluble organic carbon; HWSOC—hot water soluble organic carbon. Values are the mean of three replicates of each sample. Values followed by different letters in columns indicate significant differences according to Tukey's test ($p < 0.05$).

The enzymatic activity in bulk soil samples after 27 days of winter wheat cultivation showed that CMC-ase and xylanase activities were not noticeably different in most of the tested soils. Only the CMC-ase activity (2.13 nmol Glu h$^{-1}$ g$^{-1}$) in soil from pots fertilized with NPK and BC52% was significantly higher in comparison to all other samples (data not shown).

## 4. Discussion

The aim of this study was to evaluate the agronomic efficiency of pre-sowing NPK fertilization mixed with four types of biochar produced from pine wood chips under different thermal regimes and containing different amounts of carbon. We tested blends of different types of biochar with the same amount of NPK fertilization. The tested mixtures were applied in dosages of 1.85 g kg$^{-1}$, containing N ~7.8%, P ~3.6% and K ~4.7%, and the majority of nutrients were easily water-soluble. These blends in dosage of about 500 kg ha$^{-1}$ can be used in place of pre-sowing mineral fertilization of fertile soils in the temperate climate European zone. The effects of these blends in comparison to NPK applied alone or to unfertilized soil on soil properties, winter wheat seedling growth, the biomass yield or the ionome of seedlings were studied in a growth chamber experiment. As expected, the blending of various types of biochar differentiated by the amount of carbon with the same dose of inorganic fertilizer elicited different responses. The tested blends of NPK with biochar containing 52%, 50%, 75% or 85% carbon had a stimulative effect on winter wheat seedling emergence, but the overall effect on germination was not statistically significant in comparison with the control. The stimulation of seedling emergence could have been related to the slower release of nutrients from the NPK blended with biochar, resulting in lower osmotic pressure, as was described by Gwenzi et al. [53]. Similarly, no statistically significant effect of applying biochar in low doses on wheat seed germination was observed by Alburquerque et al. [54]. However, Solaiman et al. [55] showed that biochar generally increased the germination of wheat at application rates ranging from 10 to 50 Mg ha$^{-1}$, although they applied biochar alone as a single nutrient source. Despite no significant impact from the tested NPK biochar blends on germination, there was a significant stimulative effect on the fresh biomass yield of wheat seedlings observed after the use of an NPK blend with biochar containing 85% C, as well as on the dry biomass yield after the use of NPK blends with biochar containing 75% or 85% C compared to the application of NPK fertilizer alone. A meta-analysis done by Ye et al. [38] also showed similar results to our study, with an average of a 15% (CI: 11%–19%) increase in yield of several grain crops after the addition of biochar along with inorganic fertilization. They concluded that biochar was as effective as fertilizers in increasing crop yields when added in combination with mineral fertilizers. However, the improvement in the growth of winter wheat seedlings in our experiment cannot be related to the nutritional value of two high carbon biochars, because the amounts added to the soil of nitrogen, phosphorus and potassium were almost equal in all four types of biochar used

for the preparation of the blends. Moreover, nutrients added to the NPK fertilizers with 1.25 g of the tested biochar delivered only about 2.5%, 1.0% and 4.6% of the total amount of N, P and K, respectively. The stimulation of seedling development by BC75% and BC85% produced at a high temperature can be explained by the fact that such types of biochar generally have high surface areas [56,57] and are good adsorbents of different ions [58]. Moreover, biochar of plant origin can lower the activity of soil urease [59], slow down the release to the soil of adsorbed ammonium [60] and have remarkable adsorption capacity of nitrate ions, which is probably what led to the longer availability of N and others nutrients from the tested blends and supported better growth of the plants.

The influence of biochar was also noticeable in the ionome status of seedlings. The most noticeable effects were observed in the case of Ca and Mg, the value of which decreased, and simultaneously, the content of Mn and Fe noticeably increased in the presence of BC75% and BC85%. The increase of the content of Mn and Fe in the seedlings may have been related to the reported solubilization of these nutrients in soils at pH values below 8 with extracts of high-temperature biochar [61].

The higher content in plants of Fe can also explain the enhanced development of the seedlings in soil enriched with a blend of NPK and biochar. Fe is involved in the synthesis of chlorophyll and is essential for the maintenance of the chloroplast structure and function [62] as well as of Mn, which is involved in the water-oxidizing enzyme system [63]. Although there was a noticeable decrease in the content of Ca and Mg in the plant tissues after applying biochar with NPK, the contents of both nutrients were at a sufficient level [64]. Similar decreases in the content of Ca and Mg and a stimulative effect on growth were reported for spinach and mustard by Zemanova et al. [65], for soybean plants [66] and for lettuce [67] after applying biochar alone or with fertilizers. Additionally, the application of nutrient-rich biochar produced from animal wastes reduced Ca in the leaves of corn [68]. These reported effects, in general, can be explained by the higher accumulation of K from biochar applied in higher doses, which is antagonistic to Mg in the translocation step from the root to the shoot, according to Ohno and Grunes [69]. Additionally, Rhodes et al. [70] reported that applying K substantially reduced the content of Ca and Mg in the leaves of sugarcane. However, the content of K in the wheat seedlings in our study was not higher after applying NPK blends with biochar than after NPK was used alone. The phenomenon of a lower content of Ca and Mg in the tested seedlings can be indirectly explained by the findings of Angst and Sohi [71], who reported a slower release of Mg than of K and P from biochar.

The tested biochar added to synthetic fertilizer in an amount equivalent to 170–287 kg of biochar carbon per ha did not significantly change the physical and chemical properties of the soil except for the composition of the organic soil matter. Negative or positive effects on soil properties from different types of biochar applied in doses equivalent to a few up to hundreds of Mg ha$^{-1}$ have been summarized in the scientific literature in several reviews, e.g., [23,38,72,73]. The impact of high quantities of biochar on soil properties described in these reviews is not comparable with our observations, due to the fact that such high doses improved plant access to soil nutrients and promoted plant growth and root structure, changed the soil acidity, soil structure, ion exchange capacity and water retention capacity.

Even the low doses of BC mixed with synthetic fertilizers that were used in this study can noticeably change the SOM status of the soil in the surface layer. The increase of the content of humic acids in the soil in the presence of BC can be attributed to the phenomenon described by Wang et al. [74] and Zhang et al. [75], who suggested that a wood biochar amendment might be a potential method to enhance humification from manure composting. Moreover, Kasozi et al. [76] found greater sorption of both catechol and HA from biochar, especially those with nanopores, i.e., biochar made at higher temperatures.

These findings suggest that the observed increase in humic acids and decrease in free phenolic compounds in the soil after the addition of biochar with NPK was the effect of the sorption of these compounds and may correspond to a faster formation of aromatic polymers on the surface of the biochar. Several phenolic compounds of low molecular weight, particularly *p*-hydroxybenzoic, vanillic, *p*-coumaric and ferulic acids, are of widespread occurrence in soils and, at certain concentrations, they may negatively influence the growth of plants [77]. High concentrations of some phenolic acids

have also been reported to impair root elongation and affect the metabolism of indole-3-acetic acid, a major auxin [78]. The decline in the amount of FPH compounds after adding biochar to the soil could be an additional factor that stimulated the development of the seedlings in our experiment, because the application of biochar to the soil resulted in the adsorption of toxic compounds of natural origin, which decreased their activity, as was described by MacKenzie and DeLuca [79], Cheng and Lehmann [80] and Pignatello et al. [81]. In addition to the influence of the tested biochar on the content of HA and FPH, there was a noticeable decrease in glomalin, a glycoprotein produced by arbuscular mycorrhizal fungi (AMF); this finding similar to what was found by Warnock et al. [82]. Additionally, Brantley et al. [66] concluded that biochar application may have improved plant access to soil nutrients by promoting plant growth and root structural features, rather than by enhancing mycorrhizal infection rates. The fact that the tested biochar added to synthetic fertilizer had no effect on the level of WSOC and HWSOC as well as on the CMC-ase and xylanase activities suggests that there were no significant changes in the microbial biomass and microbial activity in the soil.

## 5. Conclusions

The pot experiment provided data to evaluate the efficacy of synthetic NPK blends with pine wood chip biochar produced under different temperature regimes. The results confirm the potential of biochar, especially those produced at a high temperature ($\geq$600 °C) and used in a low dose to improve the growth and development of winter wheat seedlings. No negative effects of the addition of biochar to NPK were observed on soil or crop quality. This confirmed that the biochar used was not a direct source of nutrients. The role of biochar in blends with NPK, which stimulated the growth of wheat seedlings, was indirect and we suggest that this effect was related to a later release of nitrogen adsorbed on the biochar, as well as an enhanced uptake of Mn and Fe by the plants. Moreover, changes in the SOM composition, the increase of HA content and the decrease of the FPH content promoted the development of winter wheat seedlings.

Considering all the observed effects and the acknowledged positive impact of biochar additives on synthetic fertilizer efficacy and on soil properties, a pre-sowing co-row soil application of biochar-blended NPK represents a promising technical option for increasing the environmental sustainability of agricultural systems. This primary agronomic evaluation provides useful information for fertilizer industry managers to integrate biochar into conventional synthetic fertilizers for agricultural practices, and for policymakers to develop measures promoting innovative technologies more environmentally friendly for use in agriculture.

**Author Contributions:** Conceptualization, G.K., R.K. and S.J.P.; methodology, G.K., R.K., E.G.M., R.Z., A.K., K.P. and S.J.P.; investigation—performed the pot experiments, G.K., R.K., E.G.M., M.P.O., J.K. and R.K.; data curation—compiled and analyzed the results, G.K. and S.J.P.; writing—original draft preparation, G.K. and S.J.P.; review and editing, G.K., R.K., E.G.M., M.P.O., J.K., R.K. and S.J.P. All authors have read and agreed to the published version of the manuscript.

**Funding:** Part of this research conducted at Częstochowa University of Technology was supported by The National Centre for Research and Development of Poland within the framework of the contract No. BIOSTRATEG G3/345940/7/NCBR/2017 (project acronym: SoilAqChar).

**Conflicts of Interest:** The authors declare absolutely no conflict of interest in the preparation and submission of this manuscript.

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
