# Peer review of "The Effect of Various Types of Biochar Mixed with Mineral Fertilization on the Development and Ionome of Winter Wheat (Triticum aestivum L.) Seedlings and Soil Properties in a Pot Experiment"

_agronomy, doi:10.3390/agronomy10121903_

Round 1

Reviewer 1 Report

Please, find in attach the comment and the requested revions

Author Response

Dear Reviewer, 

Thanks for your comments, helpful suggestions, and notation of our faults in the manuscript.

#1 Introduction. However, no informations are reported about the biochar performance in pot experiments. Please report the following: 10.3390/app10051618.

Done, The appropriate correction was done and additional reference was added in  line 92. 

#2 The abstract ...  the treatments used for the experiment were not well explained. Please add this information. Moreover, the information about the used biomasses for the obtained biochar should be added. 

Done. Instead of doses expressed as equivalent of kg ha-1 the dose of tested blends applied per pot was also described.

#3 Keywords: delete biochar since this word is also reported into the title

Done. The noun biochar was deleted.

#4 Table 1: the “biochar type I” characteristics are not reported please add
the information. Abbreviations (BC50%....) I guess it is referred to the C content, please specify.

Done by adding the description of "biochar type I" and appropriate explanations of abbreviation were added in Table 1. 

#5 Line 110: please add information about the pot material and dimension (e.g. height, size, circumference, volume)

Done. Pieces of information about the material, height, volume, and circumference of pots were added. 

#6 Line 122: please add the number of replicates for each sample. Moreover is important to specify the space among pots.

Done. The space among pots was specified. The number of replications is indicated in paragraph 2.5 to avoid several similar statements in each method.

#7 Figure 1: specify about bars meaning

Done. The bar meaning was added. 

#8 Table 2: the unit measure for nitrogen is referred also for the other macro and
microelements? Moreover the caption should be complete as reported in table 1.

Done. The unit measure was placed in the center  The captions were added to all tables. 

#9 The same for table 3

Done.  As above. 

Reviewer 2 Report

The manuscript "The Effect of Various Types of Biochar Mixed with Mineral Fertilization on the Development and Ionome of Winter Wheat Seedlings and Soil Properties in a Pot Experiment" highlighted the effect of biochar in combination with synthetic fertilizer.

The experiemnt is well designed, written comprehensively and would be a great contribution in soil research.

I have only following minor comments

  1. would be better if author can present the detailed physiochemical analysis of biochar and soil.
  2. why pots were irrigated to the level of 60% of pot water holding capcity? its a moderated water shortage stress? why not pot were well irrigated?

Author Response

Dear Reviewer, 

Thanks for your helpful remarks related to our manuscript. 

#1. Would be better if author can present the detailed physiochemical analysis of biochar and soil.

The additional data describing the biochar properties (porosity - line  111 - 113) and the soil properties (granulometric fractions - line 122 - 124) were added in paragraph 2.1. and 2.2. respectively. 

#2. Why pots were irrigated to the level of 60% of pot water holding capacity? its a moderated water shortage stress? why not pot was well irrigated?

Pots were irrigated to the level of 60% WHC because it is a standard procedure for pot experiments suggested by the Polish Institute of Soil Science and Plant Cultivation to study the efficacy of mineral fertilization. Such a level of irrigation secures a proper proportion of air-water phases. An additional explanation was added in paragraph 2.2 (line 141).